# Validation of Fresnel–Kirchhoff Integral Method for the Study of Volume Dielectric Bodies

**Soumia Imane Taleb, Cristian Neipp \*** , **Jorge Francés** , **Andrés Márquez** , **Mariela L. Alvarez,**
**Antonio Hernández, Sergi Gallego** and **Augusto Beléndez**

I.U. Física Aplicada a las Ciencias y las Tecnologías, Universidad de Alicante, P.O. Box 99, 03080 Alicante, Spain;
yousfi051291@gmail.com (S.I.T.); jfmonllor@ua.es (J.F.); andres.marquez@ua.es (A.M.);
mariela.alvarez@ua.es (M.L.A.); ahernandez@ua.es (A.H.); sergi.gallego@ua.es (S.G.); a.belendez@ua.es (A.B.)
\* Correspondence: cristian@ua.es

**Abstract:** In this work, we test a nondestructive optical method based on the Fresnel–Kirchhoff integral, which could be applied to different fields of engineering, such as detection of small cracks in structures, determination of dimensions for small components, analysis of composition of materials, etc. The basic idea is to apply the Fresnel–Kirchhoff integral method to the study of the properties of small-volume dielectric objects. In this work, we study the validity of this method. To do this, the results obtained by using this technique were compared to those obtained by rigorously solving the Helmholtz equation for a dielectric cylinder of circular cross-section. As an example of the precision of the method, the Fresnel–Kirchhoff integral method was applied to obtain the refractive index of a hair by fitting the theoretical curve to the experimental results of the diffraction pattern of the hair measured with a CCD camera. In a same manner, the method also was applied to obtain the dimensions of a crack artificially created in a piece of plastic.

**Keywords:** scattering; Fresnel diffraction; interferometry; solid mechanics; optical methods

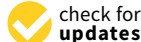



## 1. Introduction

There is no doubt that optical methods have been applied with success in different fields of engineering such as experimental solid mechanics, fracture mechanics, civil engineering, etc. Fiber-optic sensor technology, for instance, has been widely used by civil engineers for performance monitoring of civil infrastructures. This is basically due to the fact that fiber-optic sensors have the advantages of small dimensions and good resolution and accuracy [1]. Digital image correlation (DIC) has also been employed for the measurement of small deformations, such as those occurring during fluid–structure interaction [2]. Moiré interferometry has also been a valuable experimental technique for the understanding of the mechanical behavior of materials and structures [3–5]. Photoelasticity [6] has also been applied to evaluate the stress and strain field around cracks [7], etc.

In this work, we introduce another technique to evaluate properties of defects, deformations, cracks, and in general any small-volume structure that can be modeled as a volume dielectric body. The method is based on the Fresnel–Kirchhoff integral [8], which in principle was intended to simulate planar structures. Nonetheless, it also has been applied to simulate volumetric dielectric structures as well [9–11]. To do this, the structure must be treated as a two-dimensional object. This is achieved by taking into account the phase accumulated by an incident plane wave after passing through the object [12]. The amplitude of the input field in the Fresnel–Kirchhoff integral is assumed to have this phase, and the calculation of the integral provides the field in the output plane. In this work, the accuracy of this method will be analyzed by comparing the results provided by the Fresnel–Kirchhoff approximation for a dielectric cylinder of circular cross-section to the results obtained by rigorously solving the Helmholtz equation.

Next, the method will be applied to obtain the refractive index of a small-volume body such as a human hair. To do this, the normalized intensity distribution of the diffraction pattern of the hair will be measured by using a CCD camera. Then, the theoretical curve will be fitted to the experimental results by the Fresnel–Kirchhoff method to show the validity of the method. Finally, to demonstrate the potential of the method, this will be applied to obtain the dimensions of a crack artificially created in a piece of plastic.

## 2. Theoretical Models

### 2.1. Fresnel Integral Method

The basic ideas of the method proposed are explained in this section. We are interested in obtaining some particular parameters, such as refractive index, size, etc. of a particular object. The parameters to be evaluated must have the ability to change the phase of the incident light. In Figure 1, a basic scheme for the situation proposed is shown. We will assume an object with a refractive index $n$ and a thickness $d$, allowing both parameters to vary inside the object. If plane B is described with coordinates $(x',y')$, and the amplitude of the incident light is $U_A$, then the amplitude of light at plane B can be calculated as $U_B = U_A \exp(iknd) = U(x',y')$, where $k$ is the wavenumber, related to the wavelength of light, $\lambda$, as $k = 2\pi/\lambda$.

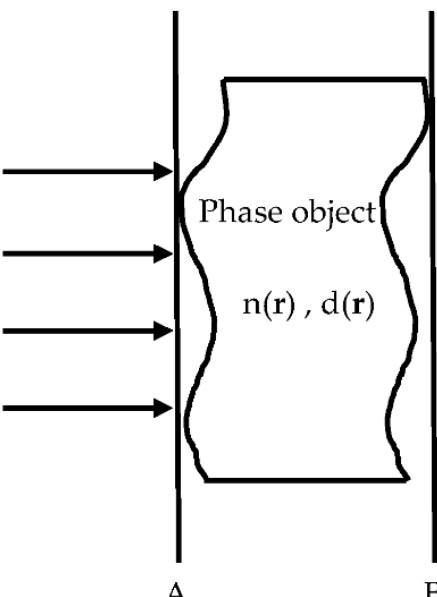

**Figure 1.** The volume object.

Once the amplitude $U(x',y')$ is evaluated at plane B, the field $U(P)$ at a point $P$ $(x,y)$ (Figure 2) can be obtained by applying the Fresnel approximation to the Fresnel–Kirchhoff integral [8]:

$$U(P) = \frac{e^{ikz}}{i\lambda z} \iint_{-\infty}^{\infty} U(x',y') \exp\left\{ i\frac{k}{2z}\left[(x-x')^2 + (y-y')^2\right] \right\} dx'dy' \tag{1}$$

The intensity of the field at point $P(x,y)$ is computed as $I(x,y) = |U(x,y)|^2$. By fitting the theoretical curve of the intensity to experimental data, one can extract information from the object under study. We will demonstrate this in Section 3. It is noticeable that in this method, the features of a volumetric object are described by a phase function, which is introduced into Equation (1). Therefore, the three-dimensional information of the object is incorporated in a two-dimensional integral. This, in our opinion, is an advantage of this method over other numerical methods existing in the literature [13], since we have reduced a three-dimensional problem to a two-dimensional one, with the correspondent saving of

computational time. On the other hand, since the method is based on the evaluation of the integral (1), which is closely related to the Fresnel transform [14], it is well suited for solving inverse problems such as in [14]. It has nonetheless some limitations; for instance, since the method makes use of the Fresnel–Kirchhoff integral, the method could be inaccurate for large values of the Fresnel number, defined in Section 3.

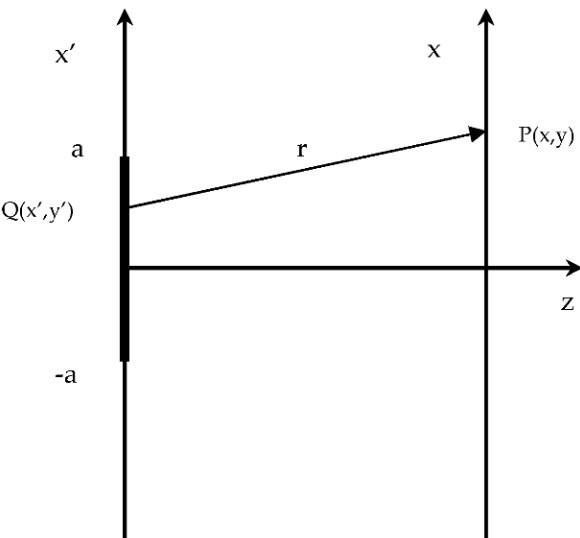

**Figure 2.** Scheme for propagation of light from the input plane to the output plane.

For the particular case of a dielectric circular cylinder, we will assume that the phases accumulated by light from a plane just in front of the cylinder, A, and another just after it, B, are [12]:

$$\varphi_a = 2ka, \qquad |x'| > a \tag{2}$$

$$\varphi_C = 2k(n_C - 1)\sqrt{a^2 - x'^2}, \ |x'| < a \tag{3}$$

where $n_c$ is the refractive index of the cylinder and $a$ is the radius of a circle representing the cross-section of the cylinder. $\varphi_a$ takes into account the phase accumulated in air, and $\varphi_c$ is the phase accumulated by the cylinder. Now, assuming that a unit amplitude wave is incident onto the cylinder, $U(x')$ can be evaluated as $U(x') = \exp(ik\varphi)$, where $\varphi$ takes into account at each point $x'$ the different contributions of $\varphi_a$ and $\varphi_C$. The expression of the amplitude of the wave field at a point $P(x,y)$ of the output plane (Figure 2) is finally:

$$U(P) = \frac{K'(z)\exp(i\varphi_a)}{B}\left\{\text{FI}(\alpha,\beta) + B\int_{-a}^{a}\exp(i\varphi_C(x'))\exp\left[ik\frac{(x-x')^2}{2z'}\right]dx'\right\} \tag{4}$$

where FI($\alpha,\beta$) depends on the Fresnel integrals of $\alpha$ and $\beta$ as:

$$\text{FI}(\alpha,\beta) = 1 + C(\alpha) - C(\beta) + i[1 + S(\alpha) - S(\beta)] \tag{5}$$

being:

$$\alpha = B(x - a) \tag{6}$$

and:

$$\beta = B(x + a) \tag{7}$$

where $B$ depends on the wavelength and the distance of the input plane to the output one:

$$B = (2/\lambda z)^{1/2} \tag{8}$$

and $K'(z)$ is:

$$K' = K \exp(ikz)/(\lambda z)^{1/2} \tag{9}$$

### 2.2. Rigorous Solution

The aim of this section is to solve Maxwell equations for the particular problem of the scattering of an infinite circular dielectric cylinder. These solutions are obtained without any approximation and will serve to test the validity of the Fresnel approximation made in Section 2.1. It is important to say that solutions of this kind exist in the literature [15,16], but in this work we have chosen another route to obtain the analytical expressions for the scattering coefficients, and we believe that this derivation is interesting on its own for the scientific community. Needless to say, although the scattering coefficients are different in this work from those obtained in other derivations, the final scattering and internal electric fields are the same.

In this derivation, the starting point is the scalar Helmholtz wave equation:

$$\nabla^2 \psi + k^2 \psi = 0 \tag{10}$$

Once scalar solutions of previous equation are obtained, vector solutions of Maxwell equations can be found in terms of the scalar solutions by building the so-called vector harmonics:

$$\vec{M} = \vec{\nabla} \times \left( \vec{e} \, \psi \right) \tag{11}$$

$$\vec{N} = \frac{\vec{\nabla} \times \vec{M}}{k} \tag{12}$$

Here, $\vec{e}$ is an arbitrary vector. In our derivation, we chose this arbitrary vector to be the unitary vector $\vec{e}_\rho$, according to Figure 3, whereas in other works [15,16] the unit vector was chosen to be $\vec{e}_z$. Although $\vec{e}_z$ is a natural choice for axial symmetric objects, we believe that the choice of $\vec{e}_\rho$ is a more general option, allowing for the simulation of dielectric bodies with other shapes. In this way, we give another expansion of the electric and magnetic fields in terms of the new vector harmonics calculated in this work (Equations (16) and (17)). In Figure 3, the axis of the cylinder under study was chosen to be the $z$ axis.

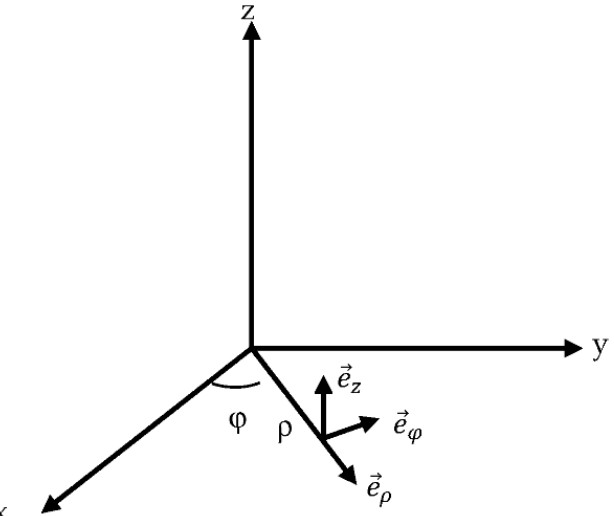

**Figure 3.** Reference frame and unit vectors for cylindrical coordinates.

From the particular configuration that we are treating here, it is clear that a proper choice of coordinates are the cylindrical coordinates ($\rho$, $\varphi$, $z$). In these coordinates, the Helmholtz equation takes the form:

$$\frac{1}{\rho}\frac{\partial}{\partial\rho}\left(\rho\frac{\partial\psi}{\partial\rho}\right) + \frac{1}{\rho^2}\frac{\partial^2\psi}{\partial\varphi^2} + \frac{\partial^2\psi}{\partial z^2} + k^2\psi = 0 \tag{13}$$

Separable solutions of this equation can be found in the form:

$$\psi_v(\rho,\varphi,z) = Z_v(k\rho)e^{iv\varphi}e^{ihz} \tag{14}$$

where $v = 0, 1, 2, \ldots$ and $h$ is dictated by the form of the incident wave. In this work, we will assume that the electric field is incident parallel to the axis of the cylinder (in the $z$ direction), so that we can consider that $h = 0$. On the other hand, $Z_v(k\rho)$ satisfy the following Bessel equation [17]:

$$\rho^2\frac{d^2}{d\rho^2}Z_v(k\rho) + \rho\frac{d}{d\rho}Z_v(k\rho) + (k^2\rho^2 - v^2)Z_v(k\rho) = 0 \tag{15}$$

For each solution of type (14), we can apply Equations (11) and (12) to calculate the corresponding vector harmonics, giving:

$$\vec{M}_v = \frac{-iv}{\rho}Z_v(k\rho)e^{iv\varphi}\vec{e}_z \tag{16}$$

$$\vec{N}_v = \frac{e^{iv\varphi}}{k\rho}\left[\frac{v^2}{\rho}Z_v(k\rho)\vec{e}_\rho + iv\left(-\frac{Z_v(k\rho)}{r} + kZ'_v(k\rho)\right)\vec{e}_\varphi\right] \tag{17}$$

where $Z'_v(k\rho) = \frac{dZ_v(k\rho)}{d(k\rho)}$; that is, the prime denotes the absolute derivative of z with respect its argument. Now the electric and magnetic fields can be expanded in terms of the cylindrical harmonics. Outside the cylinder, the electric and magnetic fields are obtained by the sum of the scattered (denoted by the subscript "*s*"), and the incident (subscript "*i*") one:

$$\vec{E}_T = \vec{E}_i + \vec{E}_s \tag{18}$$

$$\vec{H}_T = \vec{H}_i + \vec{H}_s \tag{19}$$

We will also denote the fields inside the cylinder with the subscript (1).

We will consider three expansions for the electric field: incident, scattered, and inside the cylinder. The proper Bessel functions $Z_v(k\rho)$ will be chosen accordingly for each case. In particular, since the electric field must be finite at the origin, the Bessel functions of the first kind $J_v(k\rho)$ will be chosen in the cases of the incident and the internal field. In the case of the scattered field, Hankel functions $H_v(k\rho)$ will be chosen as the generating functions, since their asymptotic behavior is that of a decaying wave at large distances. The expansion of the electric field is:

$$\vec{E}_j = \sum_{v=-\infty}^{\infty} A_v^{(j)}\vec{M}_v^{(j)} + B_v^{(j)}\vec{N}_v^{(j)} \tag{20}$$

where $j = i, 1, s$ for the incident, internal, and scattered fields, respectively.

The corresponding magnetic fields are:

$$\vec{H}_j = \frac{-ik_j}{\omega\mu}\sum_{v=-\infty}^{\infty} A_v^{(j)}\vec{N}_v + B_v^{(j)}\vec{M}_v \tag{21}$$

In this particular work, we will assume that light is polarized in the $z$ axis direction. For this particular case, it is easy to see that the incident electric field depends only on the

$\vec{M}_v$ harmonics and the magnetic field on the $\vec{N}_v$ harmonics. So, the electric field expansion for the incident light is:

$$\vec{E}_i = \sum_{v=-\infty}^{\infty} A_v^{(i)} \vec{M}_v^{(i)} \tag{22}$$

and the expansion for the incident magnetic field is:

$$\vec{H}_i = \frac{-ik}{\omega\mu} \sum_{v=-\infty}^{\infty} A_v^{(i)} \vec{N}_v \tag{23}$$

On the other hand, assuming a unit incident plane wave, the incident electric field can also be expressed in the form:

$$\vec{E}_i = e^{-ik\rho\cos\varphi} \vec{e}_z \tag{24}$$

Now, making use of the expansion of the exponential function in terms of Bessel functions [17]:

$$e^{-ik\rho\cos\varphi} = \sum_{v=-\infty}^{\infty} (-i)^v J_v(k\rho) e^{iv\varphi} \tag{25}$$

So, comparing Equations (22) and (24) and making use of Equations (16) and (25), the expansion coefficients for $E_i$ and $H_i$ can be calculated as:

$$A_v^{(i)} = \frac{\rho(-i)^{v-1}}{v} \tag{26}$$

For the case of the scattered and internal fields, the values of $A_v^{(j)}$ and $B_v^{(j)}$ are obtained imposing boundary conditions on the surface of the cylinder ($\rho = a$):

$$\left(\vec{E}_T - \vec{E}_1\right) \times \vec{n} = 0 \tag{27}$$

$$\left(\vec{H}_T - \vec{H}_1\right) \times \vec{n} = 0 \tag{28}$$

where $\vec{n}$ is a unit vector perpendicular to the surface of the cylinder and directing outward, which in this case coincides with $\vec{e}_\rho$. This is equivalent to saying that the $z$ and $\rho$ components of the electric and magnetic fields are continuous at the surface of the cylinder. Since in this work we are interested in the field outside the cylinder, we give the results obtained for the expansion coefficients for the scattered field:

$$A_v^{(s)} = -\frac{\rho(-i)^{v-1}}{v} \frac{J_v(nx)\left[J_v(x)(n-1) - nxJ_v'(x)\right] + J_v(x)nxJ_v'(nx)}{J_v(nx)\left[H_v(x)(n-1) - nxH_v'(x)\right] + H_v(x)nxJ_v'(nx)} \tag{29}$$

$$B_v^{(s)} = 0 \tag{30}$$

where $x = ka$ and $n$ is the refractive index of the cylinder.

By using Equations (16), (20), (29), and (30), the scattered field of an infinite dielectric cylinder can be obtained when a plane wave is incident perpendicular to the axis of the cylinder. Finally, with the aid of Equations (18) and (24), the total electric field at any point of the space outside the cylinder can be calculated.

## 3. Results and Discussion

### 3.1. Validation of the Fresnel Method by Comparison with the Rigorous Solution

The aim of this section is the validation of the more general Fresnel method for volume objects described in Section 2.1, which can be applied to a great number of situations, by the comparison with the rigorous solution for the dielectric cylinder obtained in Section 2.2.

To make a proper comparison, the parameters in the formalism of Section 2.2 must be addressed adequately. We present in Figure 4 the geometric scheme we considered. In this case, we want to obtain the intensity pattern created by a cylinder when light impinges on it perpendicularly to its axis on a screen that is positioned at a distance of $z_p$ from the cylinder. For the case of the rigorous solution, the intensity is obtained as:

$$I = E_T^2 \tag{31}$$

where $E_T$ is the modulus of the total electric field of Equation (20).

The intensity for the Fresnel approach is obtained from Equation (4), and multiplying the amplitude by its complex conjugate.

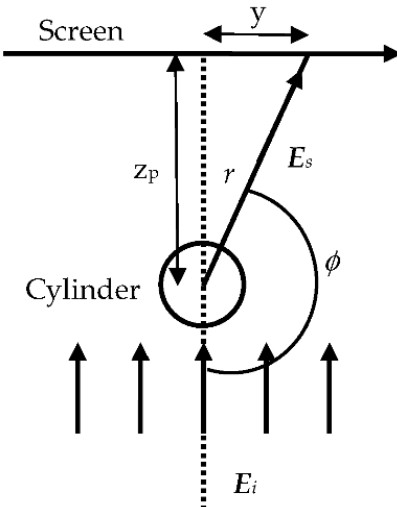

**Figure 4.** Geometric scheme for the rigorous method.

Figures 5 and 6 show the comparison of both theories for a circular cylinder. The intensity pattern was calculated at a screen that was positioned at an axial distance that was varied in the simulations. The wavelength of the incident light was chosen to be 633 nm, while the radius of the cylinder was considered as 30 μm in the case of Figure 5 and 80 μm in the case of Figure 6. It can be observed that the behavior of both curves, one obtained by using the Fresnel approximation and the other by using the rigorous solution, behaved in the same manner as a function of the axial distance. Both models gave basically the same results, but it was clear that the higher the distance to the screen, the better the agreement between both models. This is due to the fact that the Fresnel approximation works better for lower values of the Fresnel number (NF) [8], which is defined as:

$$NF = \frac{a^2}{\lambda z} \tag{32}$$

where $a$ is dimension parameter of the object under study (the radius of the cylinder in this case), $\lambda$ is the wavelength of light, and $z$ is the distance to the screen. Increasing values of $z$ give lower values of *NF*, and therefore a better behavior for the Fresnel approximation. The increase of the refractive index also worsens the results of Fresnel method slightly, which can be observed when comparing parts (a) and (b) of Figures 5 and 6. Here, part (a) corresponds to a refractive index $n = 1.85$, whereas (b) corresponds to $n = 3.4$, which is considerably higher.

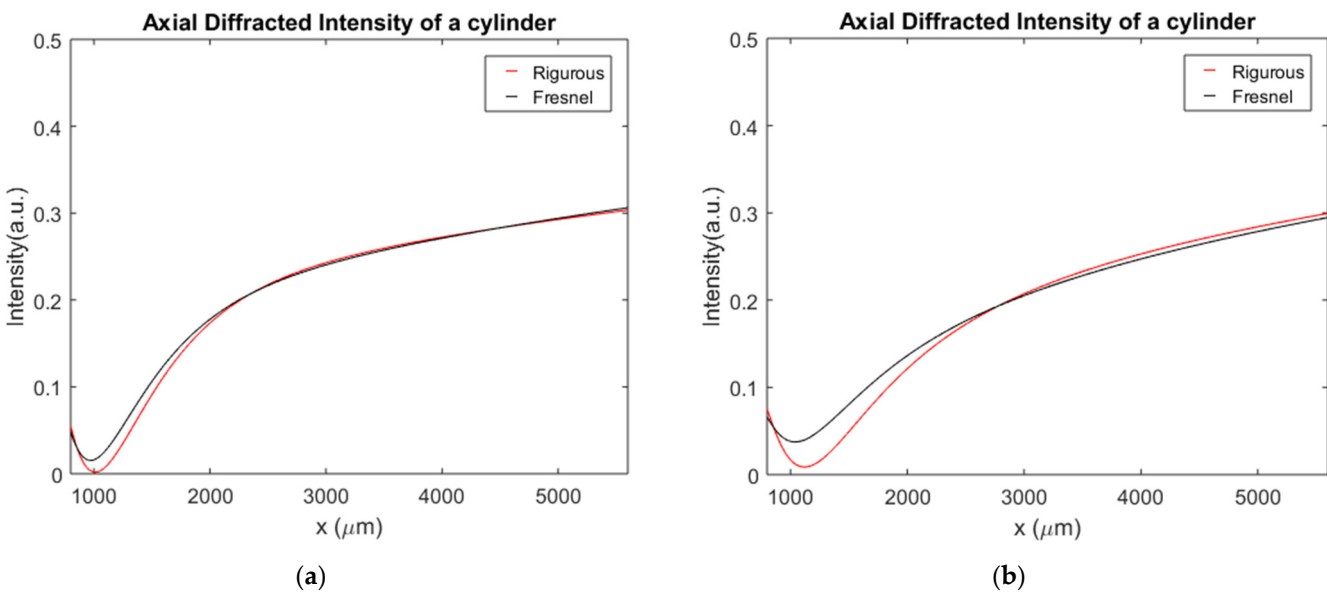

**Figure 5.** Intensity of the diffraction pattern as a function of the axial distance for a dielectric cylinder with internal radius $a = 30$ μm: (**a**) with refractive index $n = 1.85$; (**b**) with refractive index $n = 3.4$.

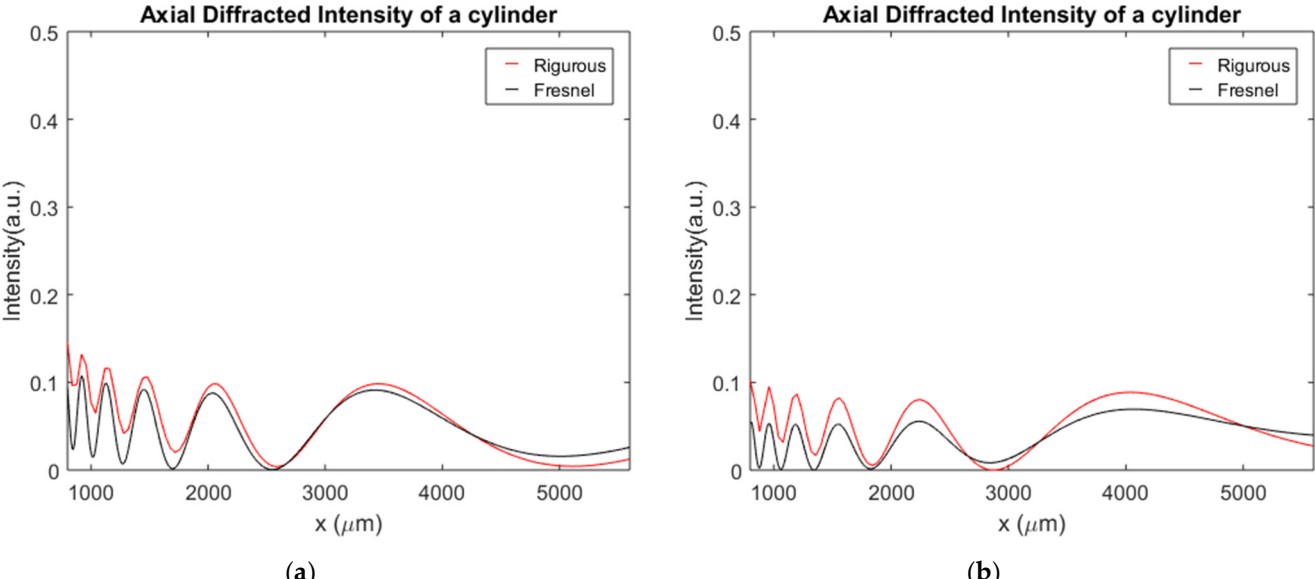

**Figure 6.** Intensity of the diffraction pattern as function of the axial distance z for a dielectric cylinder with internal radius $a = 80$ μm: (**a**) with refractive index $n = 1.85$; (**b**) with refractive index $n = 3.4$.

Despite the slight disagreement observed for both methods in the range of values considered, it must be said that the distances considered in the simulations of Figures 5 and 6 were rather conservative, since typical measuring distances from the object to the camera (CCD) were higher than 6 mm, which was the maximum distance considered in the simulations. Figures 7 and 8 show the diffraction pattern observed at a screen situated 50 mm from the cylinder for a radius of 30 μm and 80 μm, respectively. The agreement of both theories was clear in this case, thus validating the method proposed to the simulation of volume dielectric bodies.

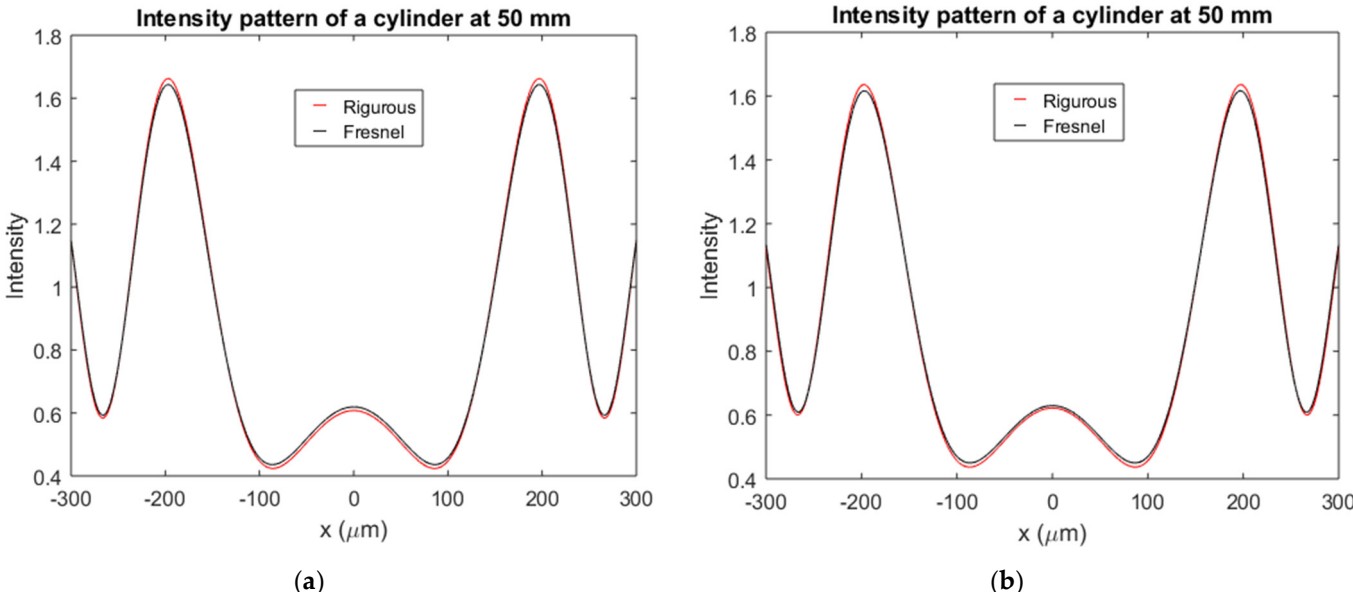

**Figure 7.** Intensity of the diffraction pattern as a function of the distance to the center of the diffraction pattern for a dielectric cylinder with internal radius *a* = 30 μm: (**a**) with refractive index *n* = 1.85; (**b**) with refractive index *n* = 3.4. The intensities were calculated assuming that the diffraction pattern was measured at an axial distance *z* = 50 mm from the fiber.

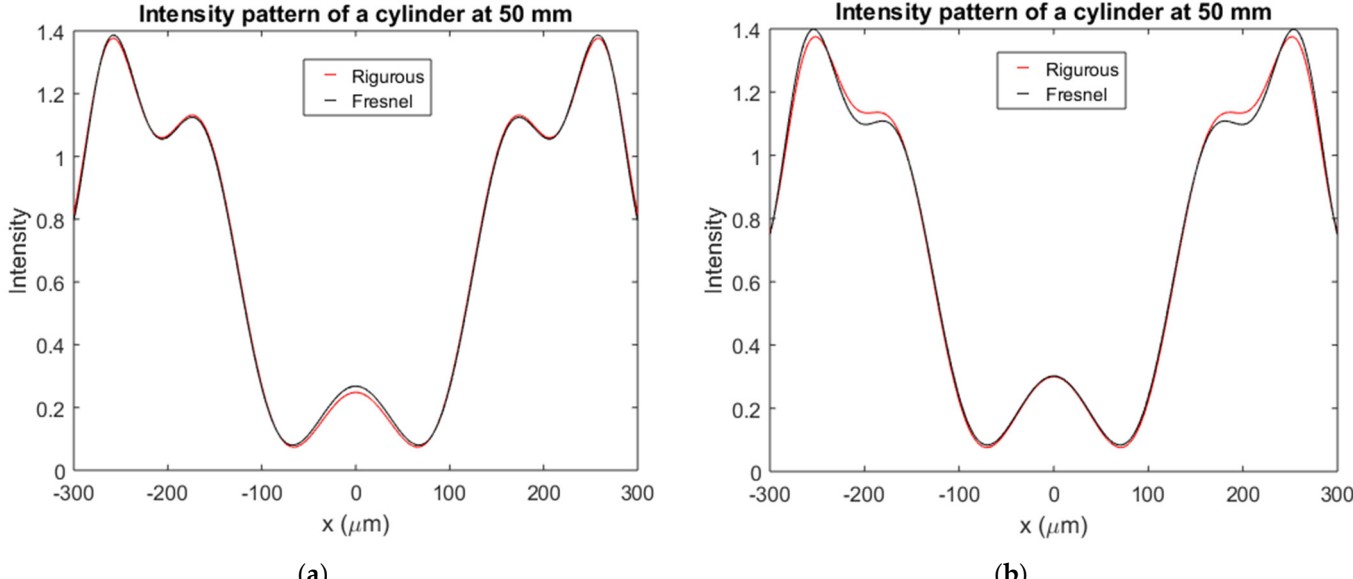

**Figure 8.** Intensity of the diffraction pattern as function of the distance to the center of the diffraction pattern for a dielectric cylinder with internal radius *a* = 80 μm: (**a**) with refractive index *n* = 1.85; (**b**) with refractive index *n* = 3.4. The intensities were calculated assuming that the diffraction pattern was measured at an axial distance *z* = 50 mm from the fiber.

This is also clear from Figure 9, where the relative error of the Fresnel method with respect to the rigorous one is depicted for a range of values for the refractive index from 1.3 to 4 and values of the radius of the cylinder from 30 to 90 μm. In order to calculate the error, the solution given by the Fresnel method and the rigorous one were calculated at each pair of values (*a*,*n*). The error was calculated at the target plane for every tested (*a*,*n*) pair using the L2-norm of the error under constant illumination. This error was divided with respect to the intensity obtained by the rigorous method integrated at the target plane. From the figure, two conclusions can be obtained: on the one hand the error in the range of

values evaluated mainly increased with the radius of the cylinder; on the other hand, it can be seen that small differences existed between the two methods.

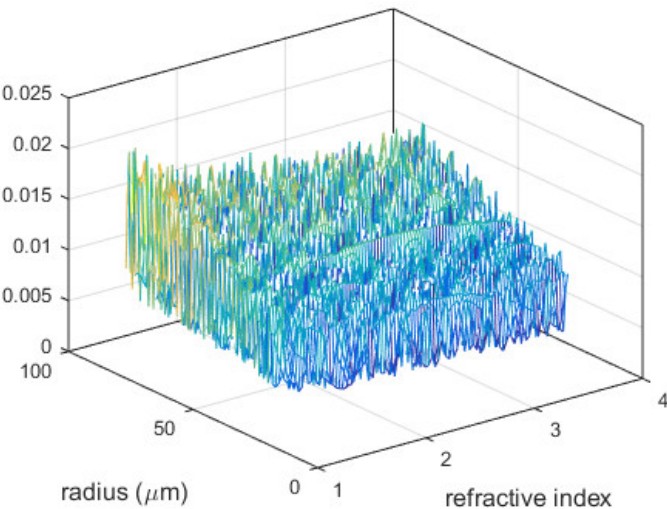

**Figure 9.** Relative error of the Fresnel method with respect to the rigorous one.

To evaluate the sensitivity of the method to changes in the refractive index and radius, simulations were carried out starting with the rigorous solution, with values for the refractive index of $n = 1.5$ and $a = 60$ μm. Then simulations were performed using the Fresnel method, slightly varying the radius and the refractive index.

Figure 10 shows the intensity profile at a distance of 20 mm from the cylinder. The rigorous solution was calculated for a refractive index of $n = 1.5$ and $a = 60$ μm, and the Fresnel method was performed for thickness values of 60, 62, 64, 66, 68, and 70 μm. From the figure, it is clear that variations of 2 μm in the radius created visible changes in the shape of the curves obtained using the Fresnel method. The distance from the maximum value to the minimum value of the curve also changed with an increase or decrease of 2 μm in the radius of the cylinder.

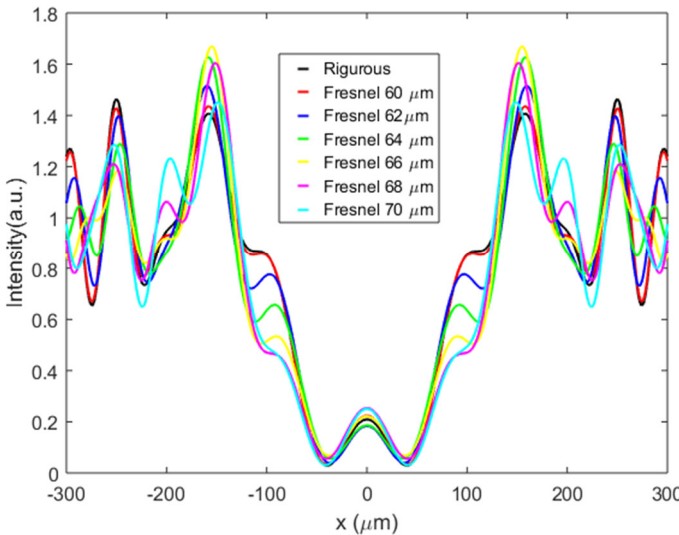

**Figure 10.** Intensity of the diffraction pattern as function of the distance to the center of the diffraction pattern for a dielectric cylinder with refractive index $n = 1.50$ and internal radii of 60, 62, 64, 66, 68, and 70 μm.

Figure 11 shows also the intensity profile at a distance of 20 mm from the cylinder, but in this case the modified parameter was the refractive index. The rigorous solution

was calculated for a refractive index of $n = 1.5$ and $a = 60$ µm, and the Fresnel method was performed for values of the refractive index of 1.50, 1.51, 1.52, 1.53, 1.54, and 1.55. As in Figure 10, variations of 0.01 created visible changes in the shape of the curves and also in the difference between the maximum and minimum value of the curve. On the other hand, it is interesting to note that small changes in the radius provoked different, although subtle, variations in the theoretical curve than in the case of changing the refractive index. For instance, if one looks the behavior of the curves in the range of x: $(-130, -70)$ µm, or symmetrically in the range of x: $(70, 130)$ µm, one can see that in the transformation of the curve at starting values (red curve) to that of the final values (light blue curve), those transforming through changes in the radius possessed a local maximum and a local minimum, whereas those transforming through changes in the refractive index did not.

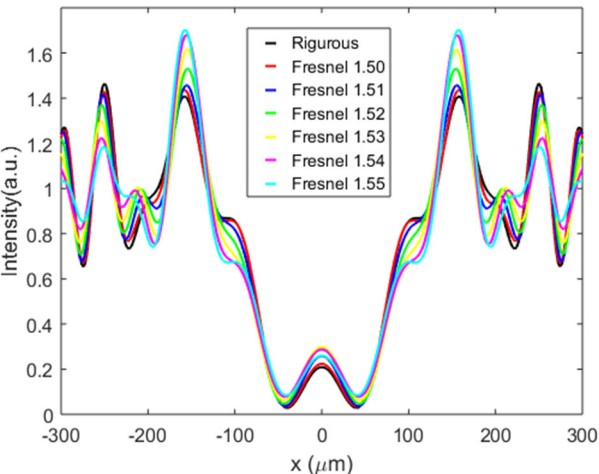

**Figure 11.** Intensity of the diffraction pattern as function of the distance to the center of the diffraction pattern for a dielectric cylinder with internal radius $a = 60$ µm and a refractive index of 1.50, 1.51, 1.52, 1.53, 1.54, 1.55.

### 3.2. Experimental Validation

Although the Fresnel method described was validated in Section 3.1, it is interesting to observe its ability to extract information from a determined volumetric dielectric object. In this section, in order to use the expressions of Section 2.1, a small cylindrical object will be studied, which in this case was chosen to be a human hair (which can be considered nearly cylindrical).

Figure 12 shows the experimental setup used to obtain the diffraction pattern of the hair. The light coming from a He-Ne laser (633 nm) was collimated by using a system of lenses; the sample (hair) was placed between the laser and a CCD connected to a personal computer, which was used to process the data. Figure 13 shows the diffraction pattern obtained from the hair by using this setup.

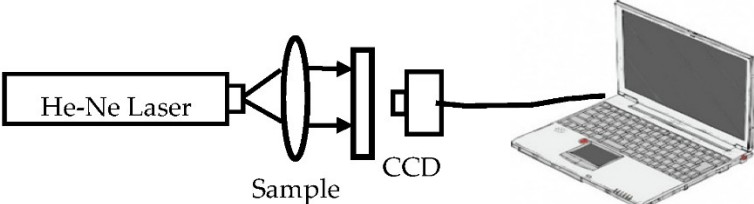

**Figure 12.** Experimental setup.

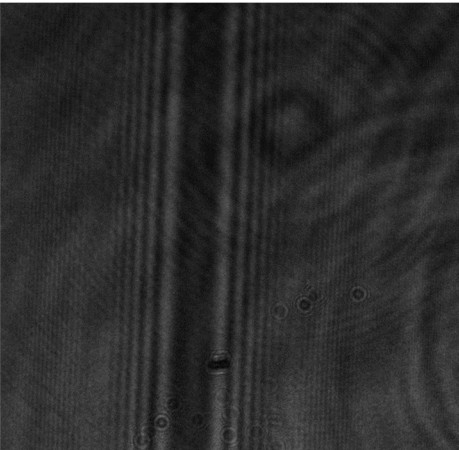

**Figure 13.** Diffraction pattern on the CCD of a hair.

In Figure 14, the normalized intensity obtained by dividing the intensity captured by the CCD by its maximum values is shown as a function of the distance to the center of the screen. In order to fit the theoretical function to the experimental data, we used the "lsqcurvefit" function of MATLAB, which implemented the "trust region reflective" algorithm [18]. The starting values for the algorithm were: radius $a$ = 50 μm and refractive index $n$ = 1.7; the fitting of the theoretical curve was obtained by using the method in Section 2.1. The experimental data provided a refractive index of 1.55 and an internal radius of 25.4 μm. The size of the hair was also measured using an optical microscope, which found a radius of 25 $\pm$ 2 μm, whereas the standard value of the refractive index of the human hair given in the literature is 1.55 [19]. The figure also demonstrates an experimental validation of the method proposed.

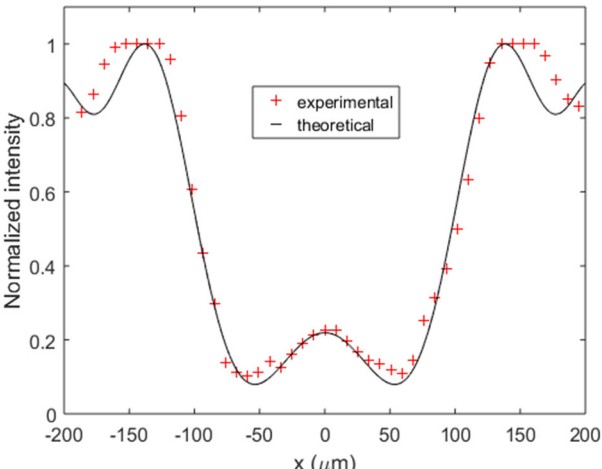

**Figure 14.** Experimental data and theoretical fit of the intensity recorded on a CCD camera from the diffraction of a hair with a refractive index of 1.55 and a radius of 25.4 μm.

Finally, in order to demonstrate the potential of the method, this was applied to obtain the dimensions of a crack artificially created in a piece of plastic. Figure 15 shows an image obtained from an optical microscope of the piece of plastic and the crack artificially created (a), and the diffraction pattern observed (b). Figure 16 shows the experimental data (extracted from the diffraction pattern) and a theoretical fit made by using the method described in Section 2.1. In this case, Equations (2) and (3) were changed to account for an elliptical crack with refractive index 1 (air), and the refractive index of the surroundings was set to 1.51 (refractive index of plastic). The varying parameter in this case was the width of the crack. As in the case of the hair, we used the "lsqcurvefit" function of MATLAB with

an initial guess of 100 µm. The width of the crack obtained with the aid of the microscope was $175 \pm 5$ µm, whereas the fit gave a value of 172 µm.

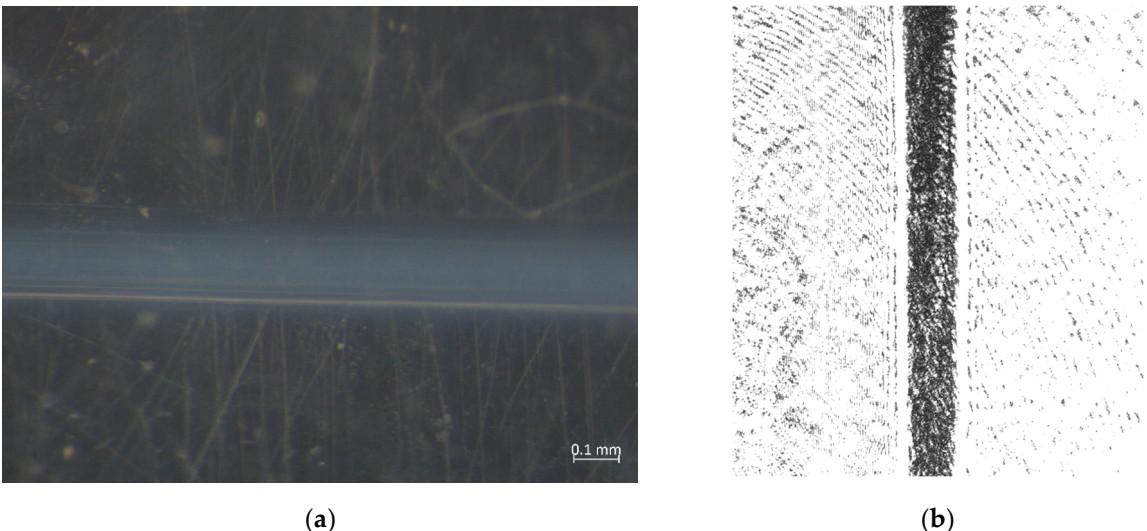

(**a**)　　　　　　　　　　　　　　　　　　　　　　　(**b**)

**Figure 15.** (**a**) Photograph obtained from an optical microscope of a crack manually generated in a piece of plastic; (**b**) diffraction pattern on the CCD of a crack in a piece of plastic.

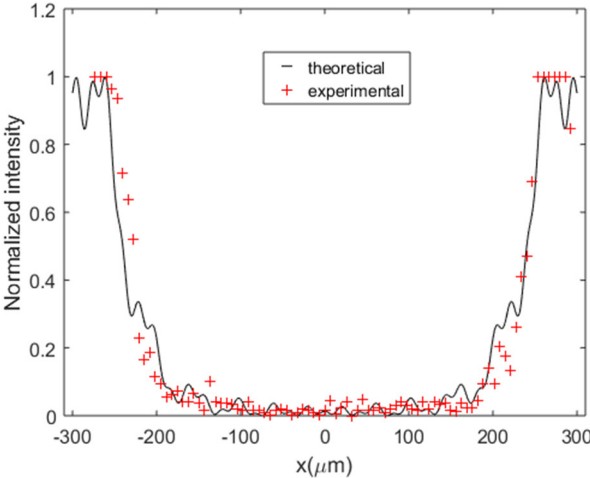

**Figure 16.** Experimental data and theoretical fit of the intensity recorded on a CCD camera from the diffraction of a crack in a piece of plastic.

## 4. Conclusions

In this work, a nondestructive optical method based on the Fresnel–Kirchhoff integral was tested. The method is suitable for volume dielectric bodies of arbitrary shapes. In particular, we studied the validity of the method by comparing it with a rigorous one for the particular case of a circular cylinder. Maxwell equations were solved exactly for this particular situation. The results demonstrated good agreement between the theories. Finally, the method was experimentally tested by observing the diffraction pattern of a human hair on a CCD camera, which also demonstrated good agreement between the theoretical model and the experimental data. To demonstrate the potential of the method, this was applied to obtain the dimensions of a crack artificially created in a piece of plastic.

**Author Contributions:** Conceptualization, S.I.T. and C.N.; methodology, S.G.; software, A.H. and M.L.A.; validation, S.G. and A.M.; formal analysis, A.B.; investigation, S.G. and C.N.; resources, A.B.; data curation, A.H.; writing—original draft preparation, S.I.T. and C.N.; writing—review and editing, C.N. and J.F.; visualization, A.M.; supervision, A.B.; project administration, A.B. and S.G.; funding acquisition, A.B. and S.G. All authors have read and agreed to the published version of the manuscript.

**Funding:** This work was supported by the "Ministerio de Ciencia e Innovación" (Spain) under project FIS2017-82919-R (MINECO/AEI/FEDER, UE).

**Conflicts of Interest:** The authors declare no conflict of interest.

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
