# Peer review of "Validation of Fresnel–Kirchhoff Integral Method for the Study of Volume Dielectric Bodies"

_applsci, doi:10.3390/app11093800_

Round 1

Reviewer 1 Report

This paper presents an optical method for studying the structure of small dielectric bodies, with apparent application to non-destructive testing. The main idea presented is quite simple and the paper is mostly well written, apart from a few typographical errors. However it makes claims, in the abstract, introduction and elsewhere, which are not supported by what is presented. These include potential to evaluate defects and small cracks, dimensions of small components and even chemical composition. None of these are actually presented in the paper and there is no indication how the technique could be applied to these sorts of features or properties.

The only application presented is the very simple case of an ideal cylinder that is later used to validate the technique. This is done two ways, analytically and experimentally. The first is by comparing diffraction pattern calculated from the solution of Maxwell’s equation (rigorous solution) to those predicted by the Fresnel method. I’m not sure why the authors have chosen to derive the rigorous solution when they admit that solutions already exist and then make reference to publications from 1957 and 1983. The tests presented are not particularly rigorous, with only two cylinder diameters and two refractive indexes examined, but the results do fit quite well. However, we have no indication of how sensitive the technique is, or how precise the results are, as all that is presented is a comparison of the patterns produced by the two methods. The authors should have included a series of results from their Fresnel method, that bracket the correct diameter and refractive index. That way we can see how sensitive it is, for all we know any solution might match just as well.

The experimental verification is done by measuring the diffraction pattern from a human hair using a HeNe laser and an optical collimator that appears to give a convergent beam, so the sample is not illuminated with plane wave light as stated earlier in the paper. This pattern is used to determine the diameter and refractive index of the hair by assuming it to be a cylinder. No attempt is made to confirm the results given, so for all we know they could be completely wrong. Why wasn’t the hair diameter measured under a microscope? What would they expect the refractive index to be? Their results do seem quite reasonable, but apparently no attempt has been made to check them. Also, why use a hair, which is inhomogeneous, rather than a glass fiber?

While this seems a reasonable idea it has not been 

Author Response

We really appreciate the comments of the reviewer. Here I explain the changes made in the manuscript corresponding to reviewer’s comments. See attachment

Reviewer 2 Report

The authors describe different derivations of diffraction theory pertaining to dielectric bodies. Although the title states "Validation of Fresnel-Kirchoff integral method for the study of volume dielectric bodies", the analysis in only limited do thin dielectric cylinders. A derivation is made using a rigorous solution, and an approximate Fresnel integral solution. This is followed by an optical experiment using a human hair.
I think there are major issues in the presentation of the manuscript, significance of the results and executed experiments.

The Fresnel integral derivation is trivial from an optics perspective. It only consists of a pure phase delay function. This is standard practice in optics for arbitrary shaped thin transmissive objects. The cylinder shape is filled in the diffraction operator in equation 4, without even solving or approximating the integral solution. This does not bring any new insights. 

No new results are obtained in the rigorous derivation, which is also stated by the authors (citations 13, 14). The authors claim that this alternate derivation may be interesting for scientific community, but do not specify why it would be interesting or how it is significantly different. The final outcome is equivalent the one found in literature as well.

The numerical experiments are insufficiently rigorous, just a few parameter combinations are selected. They are OK for illustration, but not for drawing general conclusions on their accuracy. A large range of values for "n" and "a" should be tested. This could be shown in a single map or set of graphs, by integrating over the error at the target plane for every tested (a,n) pair using for example the L2-norm of the error under constant illumination. It's incorrect to use intensity for comparing values, as this does not capture phase errors. It would be better to measure the error as the magnitude of the complex-valued difference. This should also obviate the need for testing different z, as the diffraction operator in free space is L2-norm-preserving for any z.

The paper presents multiple apparently disparate topics, thereby lacking focus. It is not clear to the reader why the Fresnel integral solution, nor the rigorous solution bring as novelty or utility. It's benefits with respect to the state-of-the-art is unclear.

No concrete applications where this validation would be applied or useful are shown, which is important for the journal "Applied Sciences". Advanced numerical algorithms already exist for the computation of diffraction in arbitrary shaped dielectric bodies.

Because of these reasons, I recommend rejection of the manuscript.

Author Response

(The authors gave the same response as above.)

Reviewer 3 Report

The authors propose an original method for calculating diffraction patterns of small objects, based on application of the Fresnel–Kirchhoff (in their transcription, Kirchoff and even Krichoff, see line 16) integral with allowance for the phase shift of light in the volume of an illuminated object. On the example of wave scattering by a thin dielectric cylinder, a good coincidence of the diffraction patterns calculated by the proposed method and obtained by a classical method of solving the wave equation is shown. On the example of laser radiation scattering by a human hair, the results of calculations are fitted to the experimental results, and good coincidence  was obtained. 

At the same time, if we assume that the purpose of fitting of the diffraction patterns is non-destructive testing, it would be desirable to comment on the possibility (or impossibility) of solving the inverse problem by this method. The computational or other advantages of this method are not disclosed.

In the description of the proposed method, the authors consider only the real part of the refractive index of the object, and the influence of losses (the imaginary part of the refractive index) on the obtained diffraction patterns is not analyzed, whereas actual objects inevitably have such losses.

The recommended relationships between the sizes of objects and sensing radiation wavelength are not discussed at al.

 In the experiment illustrated by Fig. 9 and in the text, the focused radiation is used, whereas in the Section “Theoretical models” (and in Figs. 1 and 4), the plane parallel radiation is considered.

In the comment to Eq. (31), ET is the modulus of the total electric field (line 170). Then why in Eq. (31) there is a complex conjugate to this quantity?

There are two different formulas (31).

The English language must be radically improved. In the text, there are many typos, starting from the well-known surname “Kirchoff” or “Krichoff” and the word “rugorously” (lines 14 and 246); the designations of the variables are sometimes in italic, and sometimes not. After the formulas followed by commas, the capital letters are used.

In lines 115–124, the paragraphs comprise one sentence each. In line 183, instead of “the” it is written “de”. In line 16, it is written “of the method the Fresnel-Krichoff integral method" and so on.

As a whole, the manuscript is of interest to the readers of the Journal, contains new results, and after major revision, can be published.

Author Response

(The authors gave the same response as above.)

Round 2

Reviewer 1 Report

I would like to thank the authors for taking on board the comments of the reviewers. However, some of my original concerns still remain. In particular, while a graph of the relative error has been provided in figure 9, the authors have not done what I suggested by including a series of results from their Fresnel method, that bracket the correct diameter and refractive index for the cylinder. These would have been easy to do and would be really useful in addition to figure 9. They would allow us to understand how the function changes close to the solution and understand the degree of precision required for a match. The graph of the relative error function has very poor image resolution and it’s hard to see its shape, but it does seem to have a lot of local minima over a gently changing surface. Without knowing pretty close starting values how did the authors find a solution without becoming stuck in a local minimum? In the comparison with the rigorous solution it’s easy, because they know the answer. But what would they do in the case of a completely unknown sample, as you might get for a practical application? From the shape of the error function I’m not convinced that finding the correct solution would be likely, so my confidence in finding the right result is low, and this would seem to be an underlying weakness of this technique.

The changes to section 3.1 on the hair are helpful, but how was the ‘fitting’ done? Also, why present the result of the fit as the radius of the hair and then give the measured diameter using a microscope? I know this is a minor point, but it is irritating for the reader. Again, I don’t know how they did the fitting, but if it did require an initial guess it would be useful to see what impact this has on the final solution. Human hairs range in diameter from about 20 to 150 microns, would the same solution be found for initial values across this whole range? Also, how does an error in the solution for the diameter affect that for the refractive index?

The additional test using the artificial crack, where the refractive index of the material was already known, didn’t give me much more confidence. Here the solution for the size of the crack was given as 20 by 50 microns but (as previously for the hair) an independent measurement isn’t provided to confirm the result. The quality of the fit presented in figure 14 isn’t that good either.

Overall, I remain unconvinced by this work. I think what has been presented is correct, but it has not been adequately tested and its application has not been adequately explained (especially as a large portion of the paper is still devoted to their rigorous solution, which I’m not convinced warrants inclusion). I’m also not sure how useful the technique would be for NDT. The authors need to find a specific application that only this type of measurement could address. That would make this work more relevant. How about evaluating joins in fibre optic cable in the field? But this would have to be supported with a much more rigorous evaluation than presented here.

Reviewer 2 Report

The authors have addressed several of the comments of the reviewer, for which I am grateful. There are still some issues in my view with the experimental validation.

Despite including a new experiment, the fit of the new experiment is not very good. It is not clear to what extent this is due to the accuracy of the method, or the deviation of the sample from being a "true" cylinder. No experimental reference data on the actual shape of the cracked sample is given.

There is also insufficient data on how the fitting was made. It is not clear which parameters were fixed, what the error bounds are on the different possible fits. What measure was minimized, which solution space was optimized over, what technique was used for fitting? Since the authors present these experiments as a validation of the technique, this should be detailed. It would also illustrate the utility of the method for measuring samples, providing details on both accuracy and precision.

Reviewer 3 Report

The authors have significantly improved their manuscript.

They considered all remarks of the reviewer and corrected the revised version of the manuscript. It now represents a logical, finished, and convincing study.

All comments of the reviewer to the text and language have also been considered by the authors except one: the last name of the famous scientist is Kirchhoff rather than Kirchoff. After this correction, I consider that the manuscript can be recommended for publication.

Round 3

Reviewer 1 Report

I thank the authors for taking on board my comments on the earlier versions of their paper. I am pleased to say the version I have just read is suitable for publication after the following minor corrections have been done.

  1. Update the abstract to mention the second example, measuring the dimension of the crack in plastic.
  2. Lines 62 and 63, remove the reference to chemical composition. This has not been demonstrated, and I’m still not convinced it is possible.
  3. Line 72, P is italic
  4. Line 95 subscript c in nc
  5. Line 97, subscript c in ɸc
  6. Equation 4, K’ is incorrectly formatted
  7. Equation 8 and 9, the square root sign is not done well. Consider alternative ways to present this.
  8. Line 167, italics needed for j=i,1,s
  9. Line 216, change “supposed to be” to “chosen to be”.
  10. Figure 9, I’d still like this to be a bit bigger if possible.
  11. Line 294, insert a comment about the subtle differences in the shapes of the curves in figures 10 and 11. These are most apparent around +/-200 microns, and are important for getting a unique solution.
  12. Line 318, point out a is radius, readers may have forgotten this and radius is used later on in the paragraph.

Reviewer 2 Report

The authors have adequately addressed the reviewers' comments.

I recommend acceptance of the manuscript.

Author Response

We are really grateful with reviewer’s comments throughout all the revision process. Thanks to her/his advices the article has greatly improved.